# Mice Treated Subcutaneously with Mouse LPS-Converted PrP^res^ or LPS Alone Showed Brain Gene Expression Profiles Characteristic of Prion Disease

**DOI:** 10.3390/vetsci8090200

**Published:** 2021-09-21

**Authors:** Dagnachew Hailemariam, Seyed Ali Goldansaz, Nathalie Daude, David S. Wishart, Burim N. Ametaj

**Affiliations:** 1Department of Agricultural, Food and Nutritional Science, University of Alberta, Edmonton, AB T6G 2P5, Canada; hailemar@ualberta.ca (D.H.); goldansaz@ualberta.ca (S.A.G.); 2Departments of Biological Sciences and Computing Science, University of Alberta, Edmonton, AB T6G 2E9, Canada; dwishart@ualberta.ca; 3Centre for Prions and Prion Folding Diseases, University of Alberta, Edmonton, AB T6G 2M8, Canada; daude@ualberta.ca

**Keywords:** mice, prion, lipopolysaccharide, brain genes, RML

## Abstract

Previously, we showed that bacterial lipopolysaccharide (LPS) converts mouse PrP^C^ protein to a beta-rich isoform (moPrP^res^) resistant to proteinase K. In this study, we aimed to test if the LPS-converted PrP^res^ is infectious and alters the expression of genes related to prion pathology in brains of terminally sick mice. Ninety female FVB/N mice at 5 weeks of age were randomly assigned to 6 groups treated subcutaneously (sc) for 6 weeks either with: (1) Saline (CTR); (2) LPS from *Escherichia coli* 0111:B4 (LPS), (3) one-time sc administration of de novo generated mouse recombinant prion protein (moPrP; 29-232) rich in beta-sheet by incubation with LPS (moPrP^res^), (4) LPS plus one-time sc injection of moPrP^res^, (5) one-time sc injection of brain homogenate from Rocky Mountain Lab (RLM) scrapie strain, and (6) LPS plus one-time sc injection of RML. Results showed that all treatments altered the expression of various genes related to prion disease and neuroinflammation starting at 11 weeks post-infection and more profoundly at the terminal stage. In conclusion, sc administration of de novo generated moPrPres, LPS, and a combination of moPrP^res^ with LPS were able to alter the expression of multiple genes typical of prion pathology and inflammation.

## 1. Introduction

Transmissible spongiform encephalopathies (TSEs), or prion diseases are a group of fatal neurodegenerative disorders that are characterized by misfolding of cellular prion protein (PrP^C^) into a pathogenic form, the scrapie prion protein (PrP^Sc^). The scrapie or the misfolded form of the protein is commonly found in the brain tissue of affected animals and humans. This led to the development of the prion-only hypothesis [1]. Then, the main challenge for the scientific community was to prove that indeed the misfolded protein (PrP^Sc^) was the real causative agent of prion disease. Despite, controversies regarding the potential role of a misfolded protein in the etiology of prion diseases, several reports during the last decade proved that brain homogenates coming from sick animals, or an in vitro generated misfolded protein might indeed cause typical prion disease in experimental animal models [2,3,4,5,6]. There is one drawback of all the reported investigations: they have used un-natural conditions to generate the infectious prions like serial rounds of protein misfolding cyclic amplification (PMCA) and long incubations periods, presence of RNA or other synthetic forms like polyriboadenylic acid, poly-anions like glucosaminoglycans (GAGs), denaturing acidic conditions, and synthetic lipids like 1-palmitoyl-2-oleoylphosphatidylglycerol (POPG) or phosphatidylethanolamine [3,6,7,8]. The main issue with all these successful in vitro investigations is that the odds of these co-factors being present in the brain microenvironment are null. It is difficult to bring together sonication, free lipids, free nucleic acids, and extremely low pH or un-naturally high concentrations of urea in the brain microenvironment to cause prion disease. Therefore, the search for potential and natural co-factors that might be involved in triggering prion disease is open. In addition, there is a need to better understand what is happening in the brain at the beginning of the prion disease and at the end-stage. More research is warranted regarding the use of ‘omics’ sciences to reveal gene signatures in the central nervous system (CNS) or other tissues as early in the disease process as possible and use those data as potential indicators or biomarkers of disease.

Indeed, various studies during the last decade have identified numerous genes differentially expressed in the brain tissue of experimental animals infected with various scrapie strain isolates. Up-regulation of 19 genes was reported in hamsters infected intraperitoneally with scrapie strain 263K [9]. A considerable number of these genes including encoding interferon-inducible protein 10 (IP-10), 2′,5′-oligo synthetase, Mx protein, IIGP protein, major histocompatibility complex classes I and II, complement, and β2-microglobulin were inducible by interferons (IFNs), suggesting that an IFN response is a possible mechanism of gene activation in scrapie. Mice infected with ME7 and RML differentially expressed genes encoding proteins involved in proteolysis, protease inhibition, cell growth and maintenance, immune response, signal transduction, cell adhesion, and molecular metabolism [10]. Moreover, intracerebral infection of C57BL/6 mice with mouse-adapted scrapie, ME7, and 79a, affected the expression of 158 genes in the brain tissue related to protease/peptidolysis, lysosomal functions, lipid-binding, defense- and immune-responses, cell communication, apoptosis, cell differentiation, regulation of cell growth and organization, hormone metabolism as well as ion and oxygen transport [11]. Besides, genes involved in cellular processes including protein folding, endosome/lysosome function, immunity, synapse function, metal ion binding, calcium regulation, and cytoskeletal function were identified in experimental mice [12]. In another study conducted in male mice [13], intraperitoneal infection with 301V BSE-infected mouse brain, altered the expression of genes involved in ubiquitin-mediated protein degradation, lysosomal function, protein folding, apoptosis, and calcium ion binding, transport, and homeostasis in the brain tissue. The expression of *Prnp* and *Sprn* decreases during prion infection while the expression of *Prnd* increases during neurodegeneration suggesting altered expression of these genes characterizes prion disease [14,15,16].

Recently, we showed that lipopolysaccharide (LPS) from *Escherichia coli* 0111:B4 is able to convert instantly and under normal pH, temperature, and natural conditions the Syrian hamster PrP protein into a beta-rich isoform and resistant to proteinase K (ShPrP^res^), very similar in physical characteristic with a prion [17]. Lipopolysaccharide also similarly converted recombinant mouse PrP into a β-rich protein (moPrP^res^; unpublished data) with the same physical features. We hypothesized that the in vitro generated moPrP^res^ might be infectious and cause neurodegenerative disease in FVB/N female mice and can trigger a gene-expression signature characteristic of prion disease. Therefore, the objectives of this study were to evaluate the effects of sc administration of moPrP^res^ alone or combined with chronic sc administration of LPS or chronic sc LPS alone in female FVB/N mice on the expression of genes related to prion pathology and inflammation (84 genes) in brain tissue. We also wanted to test whether chronic sc LPS would exacerbate prion disease in Rocky Mountain Laboratory-infected mice.

## 2. Materials and Methods

### 2.1. Animals and Experimental Design

The FVB/N female mice were housed in groups of 5 per cage under a 12-h light/dark cycle where food and water were provided ad libitum. All the protocols used during the study were in accordance with the Canadian Council on Animal Care [18] and were approved by the Animal Care and Use Committee, Health Sciences at the University of Alberta.

After one week of adaptation time, FVB/N mice at 5 weeks of age were randomly assigned to 6 treatment groups: (1) six weeks sc administration of saline at 11 μL/h, (2) six weeks sc injection of lipopolysaccharide from *Escherichia coli* 0111:B4 LPS at 0.1 μg/g BW at 11 μL/h, (3) one-time sc injection of de novo generated mouse recombinant prion protein (moPrP; 29-232) rich in β-sheet by incubation with LPS (moPrP^res^) at 45 μg/mouse, (4) six weeks sc injection of LPS at 0.1 μg/g BW at 11 μL/h plus one-time sc injection of moPrP^res^ at 45 μg/mouse, (5) one-time sc injection of brain homogenate from Rocky Mountain Laboratory scrapie strain at 10^7^ ID 50 units of scrapie prions, and (6) six weeks sc administration of LPS plus one-time sc injection of RML at 10^7^ ID 50 units of scrapie prions. For the groups that involve saline or LPS sc administration, ALZET^®^ osmotic minipumps (ALZET, Cupertino, CA, USA) were used. The moPrP^res^ and RML were subcutaneously injected at the start of the experiment, right after ALZET^®^ osmotic mini pumps implantation.

The ALZET^®^ osmotic pumps were implanted sc to mimic the continuous but minimal access of LPS to the body through the oral route. The sc infusion with ALZET^®^ osmotic pumps was done for 6 weeks continuously to eliminate the need for frequent animal handling and repetitive injection schedules. The sc infusion was performed according to the manufacturer’s procedure. Briefly, mice were anesthetized by isoflurane (Baxter Corporation, Mississauga, ON, Canada) and once the mouse was not responsive to tail pinch, low and continuous isoflurane, and oxygen were supplied using the anesthetic machine (Matrx by Midmark corporation, Versailles, OH, USA). After shaving the hair and disinfection, a cut was made at the back of the mice using sterile surgical scissors. The ALZET^®^ osmotic pumps were inserted and the opening was closed with sutures (ALZET Osmotic Pumps, Cupertino, CA, USA). After 6 weeks, the same procedure was repeated to take the empty pumps out and close the skin opening with sutures.

### 2.2. Euthanasia at 11 Weeks Post-Inoculation and Terminal Stage

A total of 30 FVB/N female mice, 5 from each treatment group were euthanized at 11 weeks of age. Mice were euthanized by overdose inhalation of isoflurane. After taking out the brain tissue from the skull it was sagittally cut into two symmetrical halves using a sterile scalpel. One half was snap-frozen and stored at −86 °C until RNA purification. The euthanasia of terminal stage mice was conducted at different time points. During the entire experimental period, mice were checked on a daily basis for the onset and progression of prion disease. Mice were euthanized when the clinical signs were consistent and progressive. Terminal stage sickness was characterized by kyphosis, ataxia, dysmetria, tremor, head tilt, tail rigidity, bradykinesia, proprioceptive deficits, stupor, loss of deep pain sensation, and loss of weight. The euthanasia and brain samples processing procedures were the same as the described for euthanasia at 11 weeks.

### 2.3. Total RNA Isolation

Total RNA was isolated from brain tissue (n = 3) using SV Total RNA Isolation System (Promega Corporation, Madison, WI, USA) following the manufacturer’s protocol. Briefly, brain samples were thawed at room temperature and 30 mg was taken for RNA isolation. Then samples were lysed in 175 μL RNA lysis buffer (RLA + BME) in sterile tubes. After adding 350 μL RNA dilution buffer, samples were heated at 70 °C for 3 min. Samples were then centrifuged for 10 min at 14,000× *g* (model 5430R, Eppendorf, Hamburg, Germany). The clear lysate was collected to a new collection 5 mL tube (Promega, Madison, WI, USA) and 200 μL (95%) ethanol (Commercial Alcohol, Winnipeg, MB, Canada) was added. After transferring the lysate to spin basket assembly (Promega, Madison, WI, USA) and centrifuged for 1 min, the lysate was washed with RNA wash solution. On-column DNA digestion was subsequently done using DNase I to remove any DNA contamination. After subsequent washing with washing buffers, total RNA was eluted with 100 μL nuclease-free water.

### 2.4. Custom-Made PCR Array

Custom profiler RT^2^ PCR array was designed to investigate the expression of 84 genes in 96-genes in 1 sample per plate PCR array format. The 96 wells of the PCR array consisted of specific primers for 84 genes, genomic DNA contamination control (GDC), reverse transcription control (RTC), and positive PCR control. Genes related to prion pathology and inflammation were selected from the literature. The gene symbol and ResSeq number for mouse were listed in an Excel sheet and submitted to SABiosciences (SABiosciences, Frederick, MD, USA). The custom profiler RT^2^ PCR array was manufactured by SAbioscience (SABiosciences, Frederick, MD, USA) and run in StepOnePlus Real-Time PCR System (Applied Biosystems, Foster City, CA, USA).

### 2.5. First Strand cDNA Synthesis and Quantitative PCR (qPCR) Assay

All RNA samples were checked for their concentration and purity (based on 260/280 nm measurement) using Nanodrop 8000 instrument (peqLab Biotechnologies GmbH, Erlangen, Germany). An equal amount of RNA (0.65 μg) from all the treatment groups were used for first strand cDNA synthesis using RT^2^ miRNA First Strand kit (SABiosciences, Frederick, MD, USA) following the manufacturer’s protocol. The first genomic DNA elimination mix was prepared by combining 0.65 μg total RNA and 2 μL of 5× gDNA elimination buffer. Then, ddH_2_O was added to a final volume of 10 μL. After gently mixing the contents with a pipette (Eppendorf, Hamburg, Germany) followed by brief centrifugation (Eppendorf, Hamburg, Germany), the mix was incubated at 42 °C for 5 min. Next, the RT cocktail was prepared by combining 4 μL of 5X RT buffer 3 (BC3), 1 μL of primer and external control mix (P2), 2 μL of RT enzyme mix 3 (RE3) and 3 μL of ddH_2_O to a total volume of 10 μL reaction for each sample. Then, 10 μL of RT cocktail was added to each 10 μL of genomic DNA elimination mix. After mixing gently with a pipette, the mix was incubated at 42 °C for 15 min and immediately heated at 95 °C for 5 min to stop the reaction. Finally, 91 μL of ddH_2_O was added to each 20 μL cDNA synthesis reaction and stored at −20 °C until analysis.

Expression profiling of selected genes in mice brain tissue was conducted using a custom RT^2^—PCR array (SABioscience, Frederic, MD, USA) following manufacturers’ protocol. The 96-well custom RT^2^-PCR array was designed to quantify a total of 84 genes related to prion pathology and inflammation. Five housekeeping genes (*B2M*, *Actb*, *Gapdh*, *Gusb*, and *HSP90ab1*), 3 reverse transcription controls (RTC), 3 positive PCR controls, and mouse genomic DNA contamination controls were included in the 96-well plate. Prior to real-time PCR profiling 91 μL DNase/RNase-free water was added to each of the 20 μL first strand cDNA products from each biological replicates in the treatment and the negative control group. A PCR master mix was prepared using the 102 μL diluted cDNA template, 1350 μL RT^2^ SYBR Green PCR master mix, and 1248 μL DNase/RNase-free water. Twenty-five microliter of this mix was distributed to each well of the 96-well plate containing sequence-specific primer sets for each gene and the respective controls. Following brief centrifugation (Eppendorf, Hamburg, Germany), the plate was loaded onto StepOnePlus Real-Time PCR System (Applied Biosystems, Darmstadt, Germany) and run with a thermal program of initial heating at 95 °C for 10 min followed by 40 cycles of 95 °C for 15 s and 60 °C for 1 min. The specificity of amplification was controlled using a melting curve generated at the end of the qRT-PCR protocol.

### 2.6. Protein Expression and Purification

The expression and purification of mouse recombinant moPrP (29-232) were prepared as per the procedure described previously [19]. Shortly, a synthetic gene including a 22-residue N-terminal fusion tag containing 6x His and a thrombin cleavage site (MGSSHHHHHHSSGLVPRGSHML) was synthesized by DNA 2.0 (Menlo Park, CA, USA). The gene was cloned into a pET15b expression vector between *XhoI* and *EcoRI* restriction sites and heat shock transformed into *Escherichia coli* strain BL21 (DE3). For expression, the transformed cells were grown in 100 mL Luria–Bertani broth plus 100 μg/mL ampicillin overnight to generate a starter culture. Between 1% and 2% of this starter culture was then used to inoculate 1 L of Luria–Bertani media (giving a starting D_600_ of 0.1). The cells were allowed to reach a D_600_ between 0.6 and 1.0 before induction with 1 mM isopropyl thiogalactoside. Twelve to eighteen hours later, the cells were harvested by centrifugation at 1600× *g* for 25 min at 4 °C. In addition, 15^N^-labeled moPrP (90–232) was also expressed and purified from M9 media (1.0 g/L) ^15^NH4 Cl) for the collection of heteronuclear NMR data. The inclusion of the 6x His tag afforded a standardized nickel affinity purification strategy for the protein construct.

### 2.7. Statistical Analysis

The mRNA expression of selected genes included in the custom PCR array was analyzed using the ∆∆CT method from the PCR array data analysis web portal (http://www.sabiosciences.com/pcrarraydataanalysis.php accessed on 15 June 2015). The software calculates average threshold cycle values for the biological replicates from both the treatment and control groups for all the genes in the custom PCR array. Those genes having >35 ct values and undetected ones were removed from the analysis. Data were normalized by correcting all comparative threshold (CT) values for the average ct value of endogenous controls present in the array. After sequential computations, the software calculates fold regulation and associated *t*-test *p*-value. Thus, fold change values >1.5 and *p* ≤ 0.1 were taken as cutoff values to identify genes that are differentially expressed in the treatment groups as compared to the saline-treated negative control group.

## 3. Results and Discussion

Previously, we showed that LPS from *E. coli* 0111:B4 instantly converts recombinant Syrian hamster and mouse prion protein (ShPrP^C^ and mPrP^C^) into a beta-rich isoform PrP^res^ resistant to proteinase K [17]. In the present study, we tested the hypothesis that the moPrP^res^, administered once sc, would cause prion disease in FVB/N female mice and alter the expression of genes related to prion disease and inflammation in the brain of mice as early as 11 weeks pi and terminal stage. Indeed, moPrP^res^ caused sickness and death in 60% of the experimental mice with clinical signs similar to prion disease and affected the expression of genes in the brain at 11 weeks pi and more profoundly at the terminal stage. Moreover, combinations of moPrP^res^ with LPS or treatment with RML, RML + LPS, and LPS alone also affected the expression of multiple genes at 11 weeks pi and at a terminal stage as will be discussed in detail below (Figure 1).

### 3.1. Subcutaneous Injection of LPS, moPrP^res^, RML and Combinations of LPS Altered Gene Expression in the Brains of Mice at 11 Weeks Post-Infection

The mRNA expression of genes related to prion disease and inflammation at 11 weeks pi were analyzed in all treatment groups including LPS, moPrP^res^, RML, and combinations of LPS treated groups in the brain tissue (Table 1). Results indicated that moPrP^res^ treated mice showed up-regulation of *Ifi2912a* and *Nos2* and down-regulation of *Apoe*, *Sprn*, and *C1qb* at 11 weeks pi. Subcutaneous administration of LPS for a period of 6 weeks down-regulated *Apoe*, *GbP4*, *Grn*, *Sod1*, Bax, and *Ccl17*. Combination of moPrP^res^ and LPS up-regulated 5 genes (*Tlr3*, *Tlr6*, *Nos2*, *Il1a*, and *Il1f10*) and down-regulated *C4b*. On the other hand, RML treatment down-regulated *C1qb*, *C4b*, *Grn*, *Anp32a*, and *Sod1*, whereas a combination of RML with LPS down-regulated *C1qb*, *Rtp4*, and *Sod1* and up-regulated *H2-T23* and *Ccl25*.

Interestingly, treatment with moPrP^res^ down-regulated *Sprn* (shadow of prion protein) at 11 weeks pi indicating its early alteration during the disease development. The product of this gene, Sho, is typically decreased during scrapie disease [14]. This suggests that *Sprn* may be used as a marker for the progression and development of prion and prion-like diseases.

Moreover, *C1qb* was commonly down-regulated by moPrP^res^, RML, and RML/LPS treatments at 11 weeks pi. The complement protein C1qb binds to a very wide range of non-self and altered-self-molecules. For instance, amyloid proteins are a group of altered-self-proteins to which C1qb binds. Mice deficient in C1qb, or temporarily depleted of complement component C3 by treatment with cobra venom factor showed longer survival following scrapie infection, confirming that interactions between prion and complement at the very early stages of infection affect the course of neurodegeneration [20].

The up-regulation of *Nos2* by moPrP^res^ and moPrP^res^/LPS treatments at 11 weeks pi signals the production of nitric oxide where its overproduction may compromise energy production in neurons and initiate the process of neurodegeneration [21]. The down-regulation of *Sod1* in LPS, RML, and RML/LPS groups may be a protective response of the host to the development of neurodegeneration. *Sod1* knockout animals show no signs of neurodegeneration while generation of transgenic animals expressing mutant Sod1 leads to motor neuron degeneration without reduction in Sod1 activity [22].

Mouse recombinant PrP^res^ and moPrP^res^/LPS treatments up-regulated *Il1a* and *Il1f10* at 11 weeks pi. Its elevation has been reported within brain lesions from patients with Alzheimer’s disease (AD) [23], Multiple Sclerosis (MS), Down’s Syndrome (DS), and HIV-associated dementia [24,25,26]. Furthermore, increased IL-1 has been detected in cerebral spinal fluid samples in MS, Parkinson’s (PD), and Creutzfeldt-Jakob disease (CJD) [23,27,28].

### 3.2. Reduction of Sprn mRNA Expression at Terminal Stage

Interestingly, the results showed that only one gene (i.e., *Sprn*–Shadoo (Sho) of prion protein) was differentially expressed by all treatments (moPrP^res^, LPS, moPrP^res^/LPS, RML, and RML/LPS) applied in this study (Table 2, Table 3 and Table 4). Down-regulation of *Sprn* was more pronounced in the group of mice treated with moPrP^res^/LPS (−3.46-fold) versus −3.20-fold in the LPS-treated mice, −3.13-fold in RML-treated mice as well as −2.39- and −1.94-fold in mice treated with moPrP^res^ or RML/LPS, respectively. Previously, Sho was described as a neuronal glycoprotein and a member of the mammalian prion protein family, including PrP^C^ and Doppel (Dpl) [29]. Recently, it was shown that there is a strong decrease of Sho protein in the brain of mice clinically ill with prion disease and that Sho has protective effects against prion disease [15,16]. Moreover, lowered levels of endogenous Sho have been shown to trace an early response of PrP^Sc^ buildup in the CNS and that Sho down-regulation is a common and typical event in neurodegeneration conditions caused by prion strain isolates [14]. Moreover, another study showed a relationship whereby the degree of down-regulation of the mature Sho protein was inversely related to concentrations of PrP^Sc^ [15]. Therefore, down-regulation of *Sprn* in mice treated with moPrP^res^, RML, LPS, and combinations of moPrP^res^ and RML with LPS supports the hypothesis that moPrP^res^ and other treatments caused prion-like disease in infected mice. The most intriguing observation was the down-regulation of *Sprn* by chronic subcutaneous LPS treatment. To the best of our knowledge, this is the first report that shows that chronic sc administration of LPS can lower *Sprn* expression in the brain tissue.

In a series of articles published recently [14,15,16,30,31], it is indicated that a typical finding during prion disease is decreased Sho and PrP^C^ in the brain of animals or humans affected by prion disease. A decrease of Sho and PrP^C^ are typical findings of prion disease and not present in other misfolding disorders like cytoplasmic accumulation of Tau in Tg(P301L)23027 mice [32], or parenchymal and vascular accumulation of Aβ and Bri peptide in TgCRND8 and TgADanPP7 mice, respectively [33,34,35]. Indeed, expression of *Prnp* also was decreased by moPrP^res^, moPrP^res^/LPS, RML, and RML/LPS treatments. These findings strongly support the clinical observations of prion neurodegenerative disease in mice under those treatments. Of note is the lack of significant effect of LPS on the *Prnp* gene, although LPS treatment numerically decreased expression of *Sprn*.

### 3.3. Decreased Expression of ApoE and Bax in moPrP^res^, moPrP^res^/LPS, and LPS Treatments

Two other important genes that are known to be associated with prion disease (i.e., *ApoE* and *Bax*) also were differentially expressed in 3 treatments involving moPrP^res^, moPrP^res^/LPS, and LPS compared with CTR mice (Table 2 and Table 3). The gene encoding apoE was down-regulated −2.53-fold by LPS, more than −3-fold by moPrP^res^, and more than −10-fold by moPrP^res^/LPS treatment. It has been widely documented that apoE is present in high concentrations in neurons following brain injury [36]. Moreover, some studies have shown that increased *ApoE* expression leads to synaptic regeneration [37]. In contrast, in *ApoE*-deficient mice, synaptic plasticity and regeneration are impaired [38]. Down-regulation of *ApoE* in our experiment suggests the contribution of *ApoE* down-regulation in the development of brain neurodegenerative disease in terminally ill mice treated with moPrP^res^, LPS, and moPrP^res^/LPS. The mechanism by which the *ApoE* gene is affected during brain neurodegeneration warrants further investigation.

Another important protein that has been shown consistently to participate in prion disease is Bax. *Bax* was down-regulated by moPrP^res^, moPrP^res^/LPS, LPS, and RML scrapie isolate treatments in our study. In a recent study, it was shown that *Bax* deletion in Tg(PrPΔ32-134) mice delays the development of clinical illness and slows down the apoptosis of cerebellar granule cells [39]. The Bax molecule resides in the cytoplasm and responds to various stimuli by migration to the mitochondria [40] where it can cause cytochrome C release [41], thereby activating apoptotic protease activating factor (Apaf)-1 dimerization and the apoptotic cascade. Bax is a powerful executioner of neurons and PrP has been shown to protect neurons from Bax-mediated cell death [42]. Our data suggest that down-regulation of Bax by these treatments might be a host response to slow down the process of neuronal cell death.

### 3.4. Atp1b1, Prkaca, Ncam1, and Sod1 Down-Regulated in all the Treatments Except moPrP^res^

There were five genes that were commonly down-regulated by moPrP^res^+/, RML, and RML/LPS treatments (i.e., *Sprn*, *Atp1b1*, *Prkaca*, *Ncam1*, and *Sod1*). On the other hand, a total of 9 genes were commonly influenced by RML and RML/LPS treatments; 7 of them were down-regulated (i.e., *Sprn*, *Atp1b1*, *Ncam1*, *Prkaca*, *Egr1*, *Anp32a*, and *Sod1*), and 2 of them were up-regulated (i.e., *Lyz2* and *Ly86*).

Down-regulation of the *Atp1b1* gene by moPrP^res^/LPS, RML, and RML/LPS treatments agrees with previous research indicating that failure of Na^+^/K^+^-ATPase (i.e., sodium/potassium pump) is implicated in the pathogenesis of several neurodegenerative disorders [43]. *Atp1b1* encodes the β1 unit of Na^+^/K^+^-ATPase, which is important in repolarization of neuronal plasma membrane after excitatory depolarization, the transmission of action potentials in neurons, and it is associated with caveolae and intracellular signal transduction events [44].

The neural cell adhesion molecule (Ncam), encoded by *Ncam1*, is a cell membrane constituent with three isoforms. Ncam1 plays significant roles in interneuronal and glia-neuronal adhesion phenomena, cell-cell recognition, development of the nervous system, synaptic plasticity, memory and learning as well as re-myelination and post-injury regeneration [45,46,47]. Therefore, down-regulation of this gene by moPrP^res^/LPS, RML, and RML/LPS suggests that multiple neuronal functions have been affected by these treatments.

These 3 treatments also differentially expressed *Prkaca* (Protein kinase, cAMP-dependent, catalytic, alpha), which is suggestive of apoptotic processes occurring in the brain. The protein kinase encoded by this gene binds cAMP and initiates phosphorylation of various downstream substrates and has been shown to function as a neuroprotective molecule with anti-apoptotic activity [48,49].

### 3.5. Down-Regulation of SOD1 in moPrP^res^, RML, and RML/LPS Treatments

Treatment of mice with moPrP^res^, RML, and RML/LPS also decreased the expression of *Sod1*. Decreased superoxide dismutase (Sod) activity has been shown previously in prion protein knockout mice and cell cultures [50,51,52,53], and a loss of Sod function is one of the proposed mechanisms of prion disease pathogenesis. Alterations in the activity and redistribution to mitochondria have been reported for *Sod1* mutants associated with genetic amyotropic lateral sclerosis in humans and mice [54] indicating that mislocalization and altered activity of the Sod enzymes can have deleterious consequences for neurons. Sod activity is lowered in 4 neurodegenerative diseases including AD, PD, ALS, and DS [55].

### 3.6. Up-Regulation of Immune Response Genes by moPrP^res^ Treatment

Other genes up-regulated by moPrP^res^ treatment were *Ifitm3*, *Myd88*, *Tlr4*, *Ccl5*, *Tnf*, *Fcgr3*, *Lyz2*, *Mdk*, *Tlr1*, and *Il1ra*. Up-regulation of genes related to innate immunity like *Tlr4*, *Myd88*, *Tnf*, *Tlr1*, and *Il1rn* were characteristic for the moPrP^res^ treatment only. Interestingly, mutations in the TLR4 have been shown to expedite prion disease [56]. Moreover, Myd88^-/-^ mice were fully affected by scrapie as wild-type mice, suggesting that protective effects of activation of TLR4 signaling do not result from direct interaction with prions [57]. On the other hand, MyD88 is a cytoplasmic adapter protein that associates as an obligate functional partner with all members of the TLR including TLR4 and interleukin-1 receptor (IL-1R) family [58,59]. Moreover, overexpression of Tnf is important because TNF stimulates the NF-κB transcription factor, which induces the expression of proinflammatory cytokines, and promotes the synthesis of neuronal survival factors such as calbindin, manganese superoxide dismutase, and anti-apoptotic Bcl-2 protein [60]. This cytokine also can stimulate microglia glutaminase to release glutamate, thus, generating excitotoxicity [61] and promoting the development of neurodegenerative diseases.

Treatment with moPrP^res^ up-regulated *Ccl5*, a gene that has been implicated in neurodegenerative diseases including AD. Elevated expression of Ccl5 in cerebral microcirculation is associated with the recruitment of immune cells [62]. It has also been demonstrated in cell culture models that Ccl5 can stimulate chemotaxis, increase nitric oxide secretion, and attenuate IL-10 and insulin growth factor (IGF)-1 production in activated microglia [63]. In addition to its chemoattractant function, Ccl5 also has a modulatory effect on microglia activation [61]. Treatment of neurons with Ccl5 has resulted in an increase in cell survival and a neuroprotective effect against the toxicity of thrombin and sodium nitroprusside [64]. It is obvious that mice treated with moPrP^res^ are responding to infection with enhanced Ccl5 gene expression to attract and activate microglia at the site of inflammation.

Of note was the overexpression of *Mdk* (Midkine) by moPrP^res^, which encodes a protein that is important in the attraction of immune cells to the site of inflammation. In line with this Mdk^-/-^ mice show significantly suppressed recruitment of inflammatory cells [65]. Midkine also plays a significant role as an anti-apoptotic factor for neuronal cells during apoptosis induced by serum starvation of neurons [66].

Although the role of *Lyz2* in brain neurodegenerative diseases is not clear, it is interesting to note that a recent report indicates that human lysozyme was able to prevent amyloid aggregation of the Aβ peptides [67]. Therefore, it is tempting to speculate that up-regulation of *Lyz2* in moPrP^res^-treated mice in our experiment might provide more lysozyme to the host to counteract the establishment of prion fibrils. However, this warrants further investigation.

Subcutaneous treatment with moPrP^res^ also up-regulated *Fcgr3*. Our results agree with previous research demonstrating up-regulation of this gene in scrapie-infected animals [10,68]. Fcγ receptors (FcγRs) for IgG are expressed on a wide variety of immune cells, linking cellular and humoral immunity. Engagement of activating FcγRs, associated with the common γ-chain, triggers effector cell responses, such as antibody-dependent cell-mediated cytotoxicity, phagocytosis, reactive oxygen production, and release of inflammatory mediators [69]. During inflammation, FcγRs play important roles in leukocyte recruitment and activation. More specifically FcγR3 mediates neutrophil tethering and adhesion in response to immune complexes during inflammation [70,71]. Up-regulation of the *Fcgr3* gene in the brain of mice treated with moPrP^res^ in our experiment suggests host responses to increase recruitment of leukocytes to the site of inflammation.

The other two genes up-regulated by moPrP^res^ treatment were *Tlr1* and *Ilr1a*. Tlr1 and Tlr2 have been shown to be primary receptors for Aβ and activation of inflammatory processes [72]. In addition, genetic deletion of *IL1ra* (also known as *IL-1rn*) in mice increases microglia activation and reduces neuronal survival in response to intracerebroventricular infusion of Aβ in human subjects [73].

### 3.7. Immune Response Genes Uniquely Regulated by LPS Treatment

Subcutaneous chronic infusion of LPS differentially expressed 19 genes. Out of these genes, only 6 were uniquely differentially expressed by LPS treatment including *Fcgr2b*, *Ccl17*, *Ccl19*, *Ccl25*, *Grn*, and *H2-T23*. Down-regulation of the *Fcgr2b* gene is in alignment with previous reports that *Fcgr2b* knockout mice are protected from neurodegeneration in AD [72]. Additionally, it has been reported that the FcγRIIb is a receptor of Aβ_1-42_. Furthermore, FcγRIIb is significantly up-regulated in the hippocampus area of the AD brains and in neuronal cells exposed to synthetic Aβ and that genetic depletion of *Fcgr2b* rescues memory impairments in an AD mouse model. Mice treated with LPS in our experiment are responding with down-regulation of *Fcgr2b* potentially to prevent neurodegenerative processes in the brain [74].

It is interesting to observe that two chemokine genes (*Ccl17* and *Ccl25*) were up-regulated, whereas *Ccl19* was down-regulated by LPS. The function of chemokines is to regulate leukocyte trafficking at the site of inflammation. In an earlier study, it was shown that mice deficient in Ccl17 lowered deposition of Aβ in the brain and were protected against neuronal loss and cognitive deficits. In addition, in absence of Ccl17, there is accelerated uptake and degradation of Aβ [75]. Interestingly, receptors for both Ccl17 and Ccl25 are in the hippocampal neurons, the part of the brain that is related to learning and memory [76]. On the other hand, Ccl19 was reported to be constitutively transcribed in the normal human CNS and that its transcription was elevated in MS patients. The locally produced Ccl19 might be involved in the maintenance of different types of immune cells in the brain. Therefore, subcutaneous treatment with LPS apparently affected the expression of brain chemokines to attract immune cells to the sites of inflammation.

Another gene down-regulated by LPS was *Grn*. This gene encodes the protein granulin. Granulin is expressed in neurons and microglia [77]. Mutations in the *Grn* have been shown to give rise to frontotemporal lobar degeneration (FTLD) [78]. Granulin inhibits TNF and promotes the up-regulation of Th2 cytokines such as IL-4, IL-10, and IL-5 [79]. Progranulin was shown to play a role in wound healing, where it triggers inflammation and increases the accumulation of neutrophils, macrophages, fibroblasts, and the formation of new blood vessels [80]. This suggests that down-regulation of *Grn* by LPS has influenced the etiopathology of the neurodegeneration process in the treated animals.

It is interesting to note that LPS and RML commonly affected 8 genes (*Sprn*, *Ncam1*, *Tlr3*, *Ache*, *Fcgr3*, *Ly86*, *Anp32a*, and *Bax*). Intriguingly, LPS differentially expressed 8 genes that have been previously reported in experimental scrapie disease in rodents [81]. This implicates LPS as a potential co-factor in prion disease and suggests that the effects of LPS on the etipathogenesis of neurodegeneration involve similar pathogenic mechanisms with scrapie agents.

## 4. Conclusions

In conclusion, data from this study indicate that moPrP^res^ generated in vitro by incubation with LPS was able to cause a gene expression signature of neurodegenerative disease with a resemblance with prion disease. For instance, moPrP down-regulated two genes typically related to the development of prion diseases like *Sprn* and Prnp. The combination of moPrP^res^ with LPS also differentially expressed genes typically related to prion disease especially *Sprn*, *Prnp*, and *Prnd*. Of interest is the signature of genes differentially expressed by chronic subcutaneous administration of LPS that commonly expressed 8 genes with RML, typical of the prion disease like down-regulation of *Sprn* and up-regulation of *Prnd*. RML and a combination of RML with LPS also affected a variety of genes that are involved in scrapie and that have been reported previously by other investigators. More research is warranted to study the potential role of chronic LPS and LPS-converted PrP^res^ in the development of neurodegenerative diseases in humans and animals.

## Figures and Tables

**Figure 1 vetsci-08-00200-f001:**
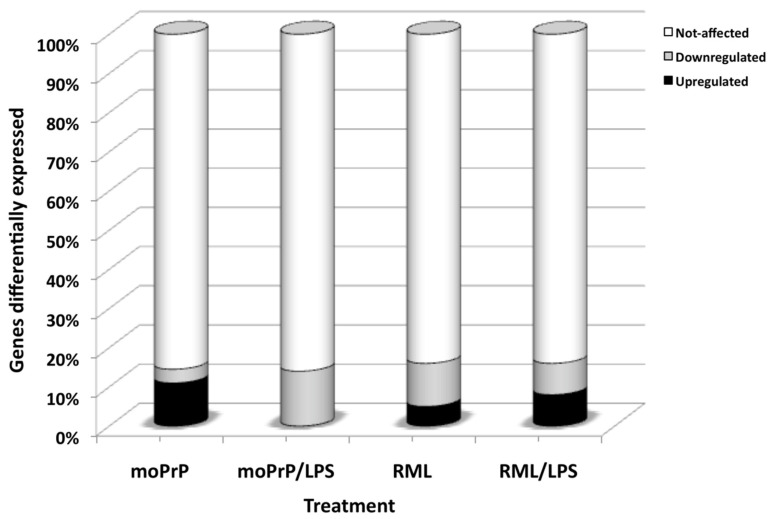
Genes differentially expressed in the brain of mice terminally sick that were euthanized after treatment with: (1) resistant mouse recombinant prion protein (moPrP^res^); (2) moPrP^res^ and lipopolysaccharide (LPS) from *Escherichia coli* 0111:B4; (3) Rocky mountain Lab brain homogenate (RML); (4) combination of RML and LPS.

**Table 1 vetsci-08-00200-t001:** List of differentially expressed genes in the brains of mice treated with LPS, moPrP^res^, moPrP^res^/LPS, RML/LPS, and RML at 11 weeks of post-inoculation.

Gene Symbol	Gene Name	Fold Change	*p*-Value
*Apoe*	Apolipoprotein	−7.6	0.001
*GbP4*	Guanylate binding protein 4	−2.4	0.02
*Grn*	Granulin	−2.4	0.02
*Sod1*	Superoxide dismutase 1, soluble	−3.0	0.001
*Bax*	BCL2-associated X protein	−3.8	0.01
*Ccl17*	Chemokine (C-C motif) ligand 17	−1.5	0.04
*Apoe*	Apolipoprotein E	−2.8	0.02
*Sprn*	Shadow of prion protein	−2.2	0.01
*C1qb*	Complement component 1, q subcomponent, beta polypeptide	−1.7	0.005
*Ifi27I2a*	Interferon, alpha-inducible protein 27 like 2A	1.8	0.01
*Nos2*	Nitric oxide synthase 2, inducible	1.5	0.04
*Tlr6*	Toll-like receptor 6	2.1	0.001
*Tlr3*	Toll-like receptor 3	2.8	0.03
*C4b*	Complement component 4B (Chido blood group)	−1.9	0.07
*Nos2*	Nitric oxide synthase 2, inducible	1.4	0.08
*Il1a*	Interleukin 1 alpha	2.2	0.06
*Il1f10*	Interleukin 1 family, member 10	1.4	0.08
*H2-T23*	Histocompatibility 2, T region locus 23	1.8	0.03
*C1qb*	Complement component 1, q subcomponent, beta polypeptide	−1.8	0.09
*Rtp4*	Receptor transporter protein 4	−2.2	0.09
*Sod1*	Superoxide dismutase 1, soluble	−1.2	0.08
*Ccl25*	Chemokine (C-C motif) ligand 25	2.3	0.01
*C1qb*	Complement component 1, q subcomponent, beta polypeptide	−1.8	0.01
*C4b*	Complement component 4B (Chido blood group)	−2.2	0.08
*Grn*	Granulin	−1.4	0.05
*Anp32a*	Acidic (leucine-rich) nuclear phosphoprotein 32 family, member	−1.9	0.05
*Sod1*	Superoxide dismutase 1, soluble	−1.5	0.01
*Bax*	BCL2-associated X protein	−1.9	0.05

**Table 2 vetsci-08-00200-t002:** List of differentially expressed genes in the brains of FVB/N mice, treated with LPS, at terminal stage.

Gene Symbol	Gene Name	Fold Change	*p*-Value
*Apoe*	Apolipoprotein E	−2.53	0.003
*Ache*	Acetylcholinesterase	−3.19	0.08
*Sprn*	Shadow of prion protein	−3.21	0.01
*Iftm3*	Interferon induced transmembrane protein 3	2.50	0.09
*Fcgr2b*	Fc receptor, IgG, low affinity IIb	−2.50	0.08
*Fcgr3*	Fc receptor, IgG, low affinity III	1.60	0.04
*Grn*	Granulin	−3.19	0.01
*H2-T23*	Histocompatibility 2, T region locus 23	1.98	0.03
*Ly86*	Lymphocyte antigen 86	1.59	0.05
*Gfap*	Glial fibrillary acidic protein	−2.02	0.03
*Anp32a*	Acidic (leucine-rich) nuclear phosphoprotein 32 family, member A	1.59	0.07
*Ncam1*	Neural cell adhesion molecule 1	3.10	0.07
*Bax*	BCL2-associated X protein	−2.00	0.04
*Ccl17*	Chemokine (C-C motif) ligand 17	−1.97	0.04
*Ccl19*	Chemokine (C-C motif) ligand 19	−2.53	0.02
*Mdk*	Midkine	4.99	0.03
*Ccl25*	Chemokine (C-C motif) ligand 25	2.49	0.04
*Tlr3*	Toll-like receptor 3	2.52	0.01
*Prnp*	Prion protein	1.27	0.55
*Prnd*	Prion protein dublet	3.15	0.06

**Table 3 vetsci-08-00200-t003:** List of differentially expressed genes in the brain tissue of terminally sick mice after subcutaneous injection of LPS-converted resistant mouse recombinant prion protein (moPrP^res^) and moPrP^res^/LPS injection.

Gene	Gene Name	Fold Change	*p*-Value
**DE genes in moPrP^res^ vs. saline**			
*Apoe*	Apolipoprotein E	−3.77	0.01
*Sprn*	Shadow of prion protein	−2.39	0.02
*Ifitm3*	Interferon induced transmembrane protein 3	4.23	0.01
*Myd88*	Myeloid differentiation primary response gene 88	1.70	0.02
*Bax*	Bcl2-associated X protein	−3.78	0.01
*Fyn*	Fyn proto-oncogene	1.69	0.02
*Tlr4*	Toll-like receptor 4	1.68	0.02
*Ccl5*	Chemokine (C-C motif) ligand 5	3.33	0.01
*Tnf*	Tumor necrosis factor	1.75	0.01
*Fcgr3*	Fc receptor, IgG, low affinity III	2.13	0.05
*Lyz2*	Lysozyme 2	1.68	0.09
*Mdk*	Midkine	2.11	0.08
*Tlr1*	Toll-like receptor 1	6.99	0.05
*Il1rn*	Interleukin 1 receptor antagonist	3.48	0.09
*Prnp*	Prion protein	−2.99	0.25
*Prnd*	Prion protein dublet	−1.17	0.98
**DE genes in moPrP^res^ + LPS vs. saline**			
*Apoe*	Apolipoprotein E	−10.98	0.01
*Sprn*	Shadow of prion protein	−3.46	0.01
*Gfap*	Glial fibrillary acidic protein	−3.45	0.03
*Atp1b1*	ATPase, Na^+^/K^+^ transporting, beta 1 polypeptide	−8.68	0.01
*Prkaca*	Protein kinase, cAMP dependent, catalytic, alpha	−3.42	0.03
*Ncam1*	Neural cell adhesion molecule 1	−4.33	0.05
*Sod1*	Superoxide dismutase 1, soluble	−3.46	0.06
*Bax*	Bcl2-associated X protein	−2.73	0.05
*Fyn*	Fyn proto-oncogene	−2.75	0.05
*Il18*	Interleukin 18	−4.35	0.06
*Adam9*	A disintegrin and metallopeptidase domain 9	−3.48	0.05
*Ache*	Acetylcholinesterase	−4.34	0.08
*Prnp*	Prion protein	−2.74	0.12
*Prnd*	Prion protein dublet	2.36	0.39

**Table 4 vetsci-08-00200-t004:** List of differentially expressed genes in brain tissue of terminally sick mice after RML and RML/LPS injection.

Gene	Gene Name	Fold Change	*p*-Value
**DE genes in RML vs. saline**			
*Sprn*	Shadow of prion protein	−3.13	0.01
*Lyz2*	Lysozyme 2	4.08	0.001
*Atp1b1*	ATPase, Na^+^/K^+^ transporting, beta 1 polypeptide	−6.20	0.01
*Ncam1*	Neural cell adhesion molecule 1	−3.90	0.02
*Prkaca*	Protein kinase, cAMP dependent, catalytic, alpha	−3.89	0.01
*Egr1*	Early growth response 1	−6.20	0.001
*Tlr3*	Toll-like receptor 3	2.56	0.03
*Ache*	Acetylcholinesterase	−3.90	0.08
*Fcgr3*	Fc receptor, IgG, low affinity III	2.59	0.09
*Ly86*	Lymphocyte antigen 86	4.08	0.06
*Anp32a*	Acidic (leucine-rich) nuclear phosphoprotein 32 family, member A	−1.95	0.08
*Sod1*	Superoxide dismutase 1, soluble	−3.12	0.06
*Bax*	Bcl2-associated X protein	−2.46	0.06
*Prnp*	Prion protein	−1.93	0.18
*Prnd*	Prion protein dublet	10.82	0.14
**DE genes in RML + LPS vs. saline**			
*Sprn*	Shadow of prion protein	−1.94	0.01
*C1qb*	complement component 1, q subcomponent, beta polypeptide	2.61	0.03
*Ly86*	Lymphocyte antigen 86	2.05	0.02
*Lyz2*	Lysozyme 2	5.18	0.01
*Atp1b1*	ATPase, Na^+^/K^+^ transporting, beta 1 polypeptide	−19.46	0.01
*Anp32a*	Acidic (leucine-rich) nuclear phosphoprotein 32 family, member A	−3.07	0.03
*Ncam1*	Neural cell adhesion molecule 1	−7.76	0.01
*Sod1*	Superoxide dismutase 1, soluble	−4.87	0.01
*Prkaca*	Protein kinase, cAMP dependent, catalytic, alpha	−3.84	0.03
*Egr1*	Early growth response 1	−4.85	0.001
*H2-k1*	Histocompatibility 2, K1, K region	2.58	0.02
*C4b*	Complement component 4B (Childo blood group)	6.51	0.09
*Ifi27i2a*	Interferon, alpha-inducible protein 27 like 2A	8.24	0.08
*A2m*	Alpha-2-macroglobulin	6.53	0.09
*Prnp*	Prion protein	−2.47	0.10
*Prnd*	Prion protein dublet	3.29	0.38

## Data Availability

All data analyzed during this study are included in this published article.

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
