# Peer review of "Mice Treated Subcutaneously with Mouse LPS-Converted PrPres or LPS Alone Showed Brain Gene Expression Profiles Characteristic of Prion Disease"

_vetsci, 2021, doi:10.3390/vetsci8090200_

Round 1

Reviewer 1 Report

Dear Editor and authors,

The manuscript ‘Mice Treated Subcutaneously with Mouse LPS-converted PrPres or LPS alone Showed Brain Gene Expression Profiles Characteristic of Prion Disease’ from Hailemariam and collaborators evaluated differential expression of genes related to TSE pathology upon subcutaneous injection of recombinant prion protein converted to a scrapie-like form by incubation with LPS. The work is interesting, but there are some points that need to be addressed/clarified before consideration for publication:

General comments:

How do you correlate the required LPS concentration to convert moPrP into a beta-sheet-rich species with the residual LPS amount eventually present in recombinant PrP preparations (as rPrP is expressed in E. coli)? The purification protocols for rPrP involve sonication of bacteria as they accumulate PrP in inclusion bodies. So, PrP is in contact with E. coli LPS during this procedure. How to explain that routinely recombinant PrP expression and purification does not generates the converted and PK-resistant PrP form?

What are the clinical signs of prion disease observed for inoculated mice in the different treatments? Histopathological analysis of brain regions would be of value to characterize whether spongiform changes and PrP deposition are present. Information on body weight changes, locomotor activity and the percent survival x incubation days (or dpi) plot are important to evaluate whether bona fide scrapie prion changes were observed.

As the authors comment that decrease of Sho and PrPC are typical findings of prion disease, I feel that measurement protein amount by western blot will be valuable for this work.

  1. Abstract: ‘converts Syrian hamster PrPC protein to a beta-rich isoform (moPrPres)’: either it is Syrian hamster (ShaPrP) or murine PrP (moPrP).
  2. It is confusing why some treatments were done with multiple sc injection in mice and other with single sc injection. Please explain.
  3. Introduction line 33: prion only hypothesis – correct to: protein only hypothesis;
  4. Do you quantify residual LPS amount in the moPrP preparation?
  5. Authors should include moPrPRes quality control (enrichment in beta-sheets and Resistance to PK digestion) prion to sc administration in mice;
  6. Line 246: the product of Sprn gene is Shadoo.

Author Response

Please see file attached for answers to Reviewers.

Reviewer 2 Report

The manuscript by Hailemariam et al. evaluates the effects of subcutaneous administration of lipopolysaccharides (LPS) converted prion protein resistant to proteinase K (PrPres) on gene expression in brain tissues. This modified protein has a very similar characteristics with a prion and thus enables to explore the pathology and inflammation associated with prion disease. The results suggest LPS as a potential co-factor in prion disease. The manuscript is interesting and original. However, the statistical analysis needs further exploration: According to the methods (line 209-219), only raw P values were obtained from differential expression analysis. The authors should make this clearer in the methods section if the P value was adjusted an how. They should ensure that the smallest familywise significance level is adjusted for multiple comparison testing. P adjust should be displayed in the tables. Based on these P values, the results should be reviewed.  Furthermore, a cutoff for P≤0.1 does not exclusively display significance in differential expression. A cutoff of 0.05 should be used.

Further comments:

Line 94-95 How many mice of which sex were used for this study?

Line 153: Please make clear what kind of selection criteria were applied to identify genes of interest in this analysis. Were novel genes and gene orthologues considered?

Line 178: the authors should state why they used these specific housekeeping genes. Have these gene been evaluated before and were they proven to be stable in expression in brain tissues in mice?

Line 365: typo: do not result…

Author Response

(The authors gave the same response as above.)

Round 2

Reviewer 1 Report

The authors have answered the main comments and critics raised by this reviewer and justified that the required data will be part of another manuscript. Additional information and results were included in the author´s response file. Thus, I now recommend the publication of the manuscript by Hailemariam and colleagues in this journal.

Author Response

Thank you. Very much appreciated. Very good questions.

Reviewer 2 Report

The manuscript has been significantly improved by the corrections. I accept the manuscript in its present form.

Author Response

(The authors gave the same response as above.)
